# Droplet superpropulsion in an energetically constrained insect

Elio J. Challita [1,2], Prateek Sehgal[1], Rodrigo Krugner [3] & M. Saad Bhamla [1] ✉

Food consumption and waste elimination are vital functions for living systems. Although how feeding impacts animal form and function has been studied for more than a century since Darwin, how its obligate partner, excretion, controls and constrains animal behavior, size, and energetics remains largely unexplored. Here we study millimeter-scale sharpshooter insects (*Cicadellidae*) that feed exclusively on a plant's xylem sap, a nutrient-deficit source (95% water). To eliminate their high-volume excreta, these insects exploit droplet superpropulsion, a phenomenon in which an elastic projectile can achieve higher velocity than the underlying actuator through temporal tuning. We combine coupled-oscillator models, computational fluid dynamics, and biophysical experiments to show that these insects temporally tune the frequency of their anal stylus to the Rayleigh frequency of their surface tension-dominated elastic drops as a single-shot resonance mechanism. Our model predicts that for these tiny insects, the superpropulsion of droplets is energetically cheaper than forming jets, enabling them to survive on an extreme energy-constrained xylem-sap diet. The principles and limits of superpropulsion outlined here can inform designs of energy-efficient self-cleaning structures and soft engines to generate ballistic motions.

Consumption of nutrients and subsequent waste elimination are hallmark functionalities of a living organism. Although fluid feeding in insects (moths, mosquitoes, leafhoppers) has received considerable attention since Darwin's time[1], little is known about the science and biofluid dynamics phenomena associated with waste elimination, despite having important ecological, morphological and evolutionary implications[2]. Specifically, we focus on how excretion influences small-bodied animals' behavior, morphology, and energetics since they face unique challenges due to their high metabolic rate[3] and physical limits set by the natural world[4]. Millimeter-sized xylem-feeding insects exemplified here with sharpshooter insects (*Cicadellidea*) face dual fluid dynamic challenges of surface tension due to their small size and energy constraints due to their xylem sap diet. Plant's xylem sap is very poor in nutritional compounds (95% water[5]) and energetically costly to pump out since it is under negative tension[6] ($\leq -1\,MPa$). To survive on this frugal diet, sharpshooter insects use large cibarial muscles and an efficient digestive system (filter chamber) to extract and filter large volumes of the xylem fluid (up to 300 × body weight/day[7,8] compared to ~1/40 × body weight/day for humans). Subsequently, sharpshooters must constantly and efficiently excrete their fluidic waste (~99% water, 'leafhopper rain'), contributing to their role as plant disease vectors[9]. Here, we ask: what are the fluidic, energetic, and biomechanical principles that enable tiny xylem sap-feeding insects to survive on a nutrient-sparse diet?

## Results

### Kinematics of droplet catapulting

Using high-speed imaging, we examine the dynamics of excretion of glassy-winged sharpshooters (GWSS, *Homalodisca vitripennis*, $n = 5$ individuals, $N = 22$ droplet ejections). Tracking the angle $\theta$ between the anal stylus and the axis along the body of the insect reveals three consecutive phases: droplet formation, spring loading, and droplet

[1]School of Chemical & Biomolecular Engineering, Georgia Institute of Technology, 311 Ferst Drive NW, Atlanta, GA 30332, USA. [2]George W. Woodruff School of Mechanical Engineering, Georgia Institute of Technology, 801 Ferst Drive NW, Atlanta, GA 30318, USA. [3]United States Department of Agriculture, Agricultural Research Service, San Joaquin Valley Agricultural Sciences Center, Parlier, CA 93648, USA. ✉e-mail: saadb@chbe.gatech.edu

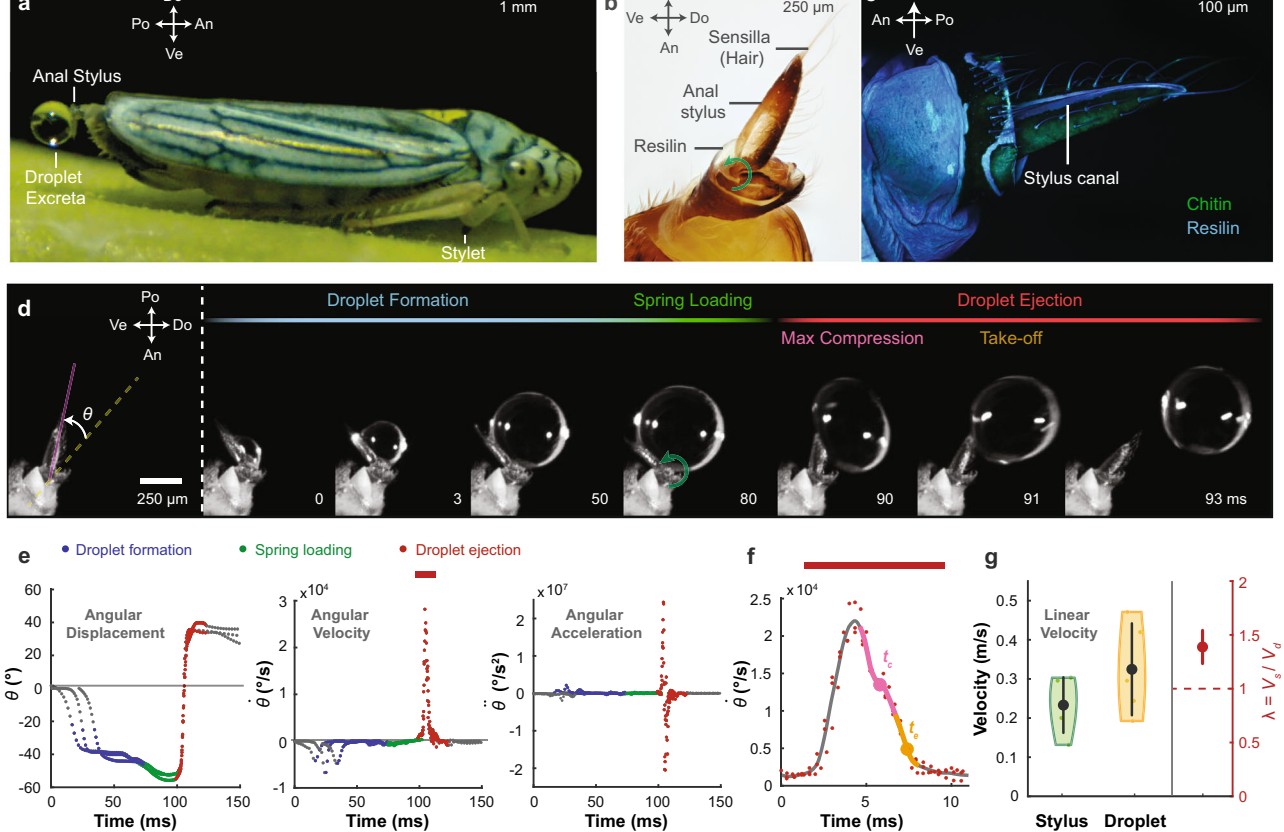

**Fig. 1 | Sharpshooter insects exhibit ultrafast droplet excreta catapulting. a** A blue-green sharpshooter (BGSS) *Graphocephala atropunctata* feeding on xylem fluid on a basil plant stem. The insect sucks fluid using its stylet before excreting the water-based waste in the form of droplets on its anal stylus (anal ligulae). **b** A close-up microscopic image of the anal stylus of a glassy-winged sharpshooter (GWSS) *Homalodisca vitripennis* reveals a clear resilin blob near the pivot point at its base and tiny hair projections (sensilla) along its surface. **c** Confocal image of the stylus (GWSS) reveals that the stylus is a complex composite of chitin (sclerotized) and resilin (flexible). A canal spans along the stylus that channels the excreted fluid during excretions. **d** Droplet excretion takes around ~100 *ms* and may be divided into three different phases. 1- Droplet formation, where a droplet excreta forms on the anal stylus. 2- Spring loading, where the insect compresses the resilin located at the anal stylus. 3- Droplet ejection, where the stylus rotates rapidly, ejecting the droplet in the dorsal and posterior plane. Before taking off, the elastic droplet undergoes compression and extension. **e** Kinematic analysis of the angle between the anal stylus and the insect's body axis (GWSS, $n = 1$ individual, $N = 3$ droplet ejections) highlighting the different excretion phases **f** Due to the elasticity mediated by the droplet's surface tension, the max compression time $t_c$ and ejection time $t_e$ occur after the stylus reaches maximum speed. **g** The max speed of the droplet at ejection $V_d$ is higher than the maximum speed of the stylus $V_s$ (GWSS, $n = 5$ individuals, $N = 22$ droplet ejections, mean of means ± STD). The speed ratio $\lambda = V_d/V_s$ is higher than unity. Po Posterior, Do Dorsal, An Anterior, Ve Ventral. Error bars represent average value ± one standard deviation.

ejection (Fig. 1d). The anal stylus starts by rotating from a neutral position to squeeze out the fluidic waste. During pumping, a water droplet ($\gamma = 72\ mN/m$, $\rho = 996\ kg/m^3$) gradually grows in the normal direction to the stylus reaching a diameter of $D_o = 725 \pm 188\ \mu m$ (Mean of mean ± SD) in a formation time ($\tau_d$) of about 80 *ms* while the stylus remains at a quasi-constant angle (Fig. 1d). As the excreted droplet approaches its final sessile state, the stylus further rotates by about ~15° (Fig. 1e), compressing and loading elastic energy into the soft resilin-structure that is surrounded by stiffer sclerotized layers (Supplementary Information, Section I, Supplementary Fig. 1 and Supplementary Fig. 3). During the droplet ejection phase, the stylus rotates rapidly, reaching a peak angular speed ($\dot{\theta}$) of $3.31 \pm 1.31 \times 10^4\ °/s$ and a peak linear speed ($V_s$) of $0.23 \pm 0.07\ m/s$. Notably, the catapulted water droplets are ~40% faster speeds ($V_d = 0.32 \pm 0.1\ m/s$) than the stylus. Calculating the speed ratio $\lambda = V_d/V_s$ reveals that $\lambda > 1$, suggesting a superpropulsive regime (Fig. 1g), which we discuss in detail in the next sections[10,11].

## Superpropulsion of liquid droplets

The physics of elastic projectiles had been previously explored in the context of throwing or hitting sports balls[12,13], kicking water droplets

from superhydrophobic surfaces[10,14], and propulsion of soft materials[11,15]. One critical feature in elastic propulsion is the importance of sequential timing and synchronization between an actuator and projectile to enhance propulsion by optimizing energy transfer[10,11,16]. Superpropulsion is a principle previously described in synthetically engineered systems, where a water droplet may be ejected at a higher speed (up to 1.6 times) than the maximum speed of a superhydrophobic vibrating plate (i.e. $\lambda > 1$)[10]. Such counterintuitive propulsion is achieved only in elastic projectiles (such as a water droplet) by carefully tuning the underlying actuator's vibrational frequency to the projectile's natural frequency. The superpropulsion principle may be exploited to enhance the throwing of rigid projectiles through the elastic tuning offered by compliant substrates[11,15]. By adding a soft layer with specific geometrical properties to a rigid projectile, the reaction force between the thrower and the projectile might be delayed to increase the initial take-off speed[11].

Here, we discover and present the superpropulsion phenomenon for the first time in a biological system, i.e., sharpshooter insects. Based on the kinematic and physical properties of the ejected droplets in sharpshooters, the Weber number $We = \rho D_o V_d^2/\gamma \sim 10^{-1}$ and Bond number $Bo = \rho g D_o^2/\gamma \sim 10^{-1}$ (where $D_o < 1\ mm$ is the diameter of the

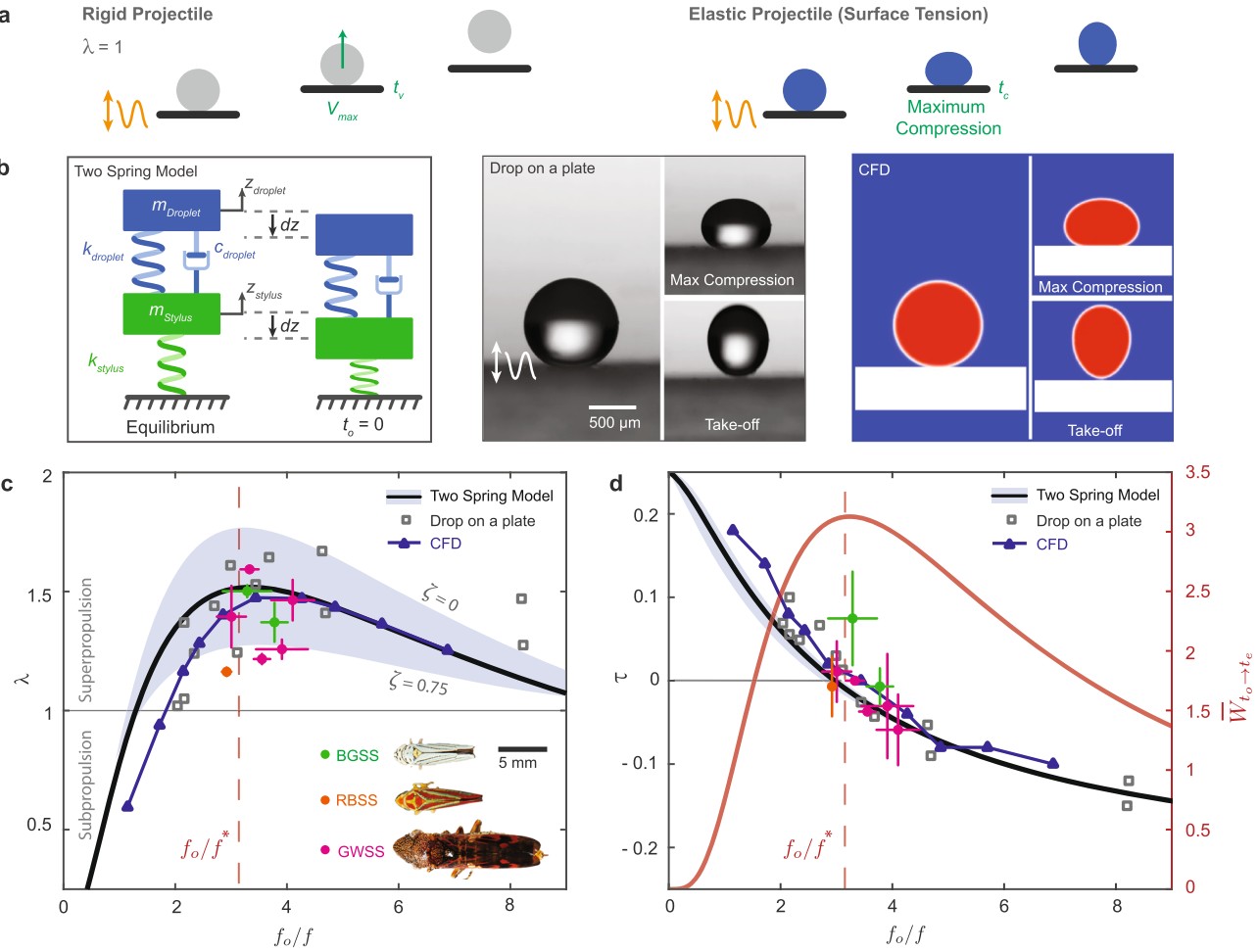

**Fig. 2 | Limits and principles of superpropulsion combining experiments, theory, and modeling. a** An ideal rigid projectile sitting on a vibrating plate will take off when the plate reaches maximum velocity $V_{max}$ at $t_v$ with an ejection speed $V_e = V_{max}$ (i.e., $\lambda = 1$). Alternatively, an elastic water droplet ($Bo < 1$) experiences deformation mediated by surface tension during ejection, reaching maximum compression at $t_c$ before ejection. **b** Droplet ejection is modeled using a reduced-order oscillator model, drop-on-plate experiment, and Computational fluid dynamic (CFD) simulations. **c** Theoretical and computational modeling predicts two separate regimes: superpropulsion ($\lambda > 1$) and subpropulsion ($\lambda < 1$). Superpropulsion occurs at $f_o/f \sim 1.16 - 1.5$ and peak kinematics around $(f_o/f)^* \sim 3.23 - 3.44$ for different damping ratio $\zeta$ in the upper system (where the shaded area represented the family of curves for different $\zeta$ values). The solid line corresponds to the

curve at $\zeta = 0.25$, and $(f_o/f)^* = 3.23$ corresponds to peak kinematics with no dissipation ($\zeta = 0$). These predictions are validated by data of three sharpshooter species and vibrating droplet-on-plate experimental data which lie the predicted region for superpropulsion. **d** Droplet dynamics reveal that peak normalized energy transfer $\bar{W}_{t_o \to t_e}$ occurs when the droplet reaches maximum compression $t_{c, max}$ matches the times the plate reaches maximum velocity $t_v$ where $\tau = (t_c - t_v)/T = 0$ at $(f_o/f)^* \sim 3.23$. For sharpshooters, the droplets' maximum compression occurs around the stylus's maximum velocity. For sharpshooters, the droplets' maximum compression occurs around the stylus's maximum velocity. Error bars of all data points in this figure represent an average value ± one standard deviation (GWSS, $n = 5$, $N = 22$; BGSS, $n = 3$, $N = 9$; RBSS, $n = 1$, $N = 3$) where $n$ = number of individuals, $N$ = number of droplet ejections.

droplet and $g$ is the gravitational constant) indicate that surface tension forces dominate the ejection dynamics over bulk forces. As a result, the droplets visibly deform into oblate spheroids reaching maximum compression at $t_c$ before extending, supporting elastic energy storage for superpropulsive principles (Fig. 1d, f, Supplementary Video I). To test the hypothesis that sharpshooters exploit superpropulsion during droplet ejection via temporal tuning between the stylus and the droplet, we apply two-spring oscillator theory, physical models, and computational fluid dynamics (CFD) and establish the principles and limits of superpropulsion. We further investigate the energetic advantages of droplet ejection via superpropulsion for the survival of these insects living on an energy-constrained xylem-sap diet.

## Superpropulsion principles in sharpshooter insects
We model the droplet ejection mechanism using a reduced-order mathematical model that consists of a 1D two-spring oscillator where

the upper mass-spring-damper ($m_d, k_d, c_d$) represents the droplet and the lower mass-spring ($m_s, k_s$) represents the stylus and resilin (Fig. 2b). Both the droplet ($We \sim 10^{-1}$, $14-28\,g$ acceleration) and resilin are modeled as an elastic Hookean spring[14,17–19] (Supplementary Information, Section II). Dissipation in sessile droplets due to viscous effects, interfacial boundary layer damping, and moving contact line[20–22] is modeled with a linear damper having a damping ratio ($\zeta$) ranging from 0 (low dissipation) to 0.75 (high dissipation). We calculate the frequencies $f_o$ and $f$, where $f_o = (1/\pi)\sqrt{k_d/m_d}$ corresponds to the second harmonic of the undamped natural frequency of the upper spring and $f = (1/2\pi)\sqrt{k_s/m_s}$ corresponds to the frequency of the lower spring. The theoretical speed ratio $\lambda_{th} = V_{d,th}/V_{s,th}$ is extracted for various values of the frequency ratio $f_o/f$ and $\zeta$, where $V_{d,th}$ is equal to the maximum speed of the upper mass and $V_{s,th}$ is equal to the maximum speed of the lower mass. The superpropulsion regime is highlighted by $\lambda_{th} > 1$ whereas the subpropulsion regime is represented by $\lambda_{th} < 1$ (and $\lambda_{th} = 1$ corresponds to a rigid-projectile-like behavior).

The two-spring model predicts a family of characteristic curves highlighted in Fig. 2c, d by the shaded area. The model predicts that superpropulsion i.e. $\lambda_{th} > 1$ occurs at $f_o/f \geq 1.16-1.5$ with a peak ranging from $\lambda_{th}^* = 1.25 - 1.76$ at frequencies $(f_o/f)^* \sim 3.23 - 3.44$, respectively. Interestingly, the two-spring model also predicts the subpropulsion ($\lambda_{th} < 1$) regime for $f_o/f < 1.16$, suggesting that at these frequency ratios, the elasticity of the projectile undermines its kinematic performance during take-off (Supplementary Information, Section III and Supplementary Fig. 8).

We compare the theoretical prediction to the field data of three different species of sharpshooter insects: Glassy-winged sharpshooter *Homalodisca vitripennis* (GWSS, $n = 5, N = 22$), blue-green sharpshooter *Graphocephala atropunctata* (BGSS, $n = 3, N = 9$) and red-banded sharpshooters *Graphocephala coccina* (RBSS, $n = 1, N = 3$). We calculate the frequency of droplet vibrations ($f_o$) as the second mode of the Rayleigh frequency $f_o = (1/2\pi)\sqrt{8\gamma/\rho R_o^3}$ [23] using high-speed footage, and the frequency of stylus ($f$) using the kinematics analysis of the stylus (Supplementary Information, Section II and Supplementary Fig. 2). Our results show that the velocity ratio $\lambda$ is $> 1$ for all species and aggregate within a small range of $f_o/f \sim 3-4$ (Fig. 2c), clearly establishing that sharpshooters exploit superpropulsion phenomenon for excretion. We supplement our results with CFD simulations and droplet-on-plate experiments that are in good agreement with both the theory and organismal data (Fig. 2b–d, Supplementary Information, Section II).

We conjecture that sharpshooters time the maximum speed of their stylus to the deformation dynamics of their droplet-excreta. To test this, we compare the time at maximum droplet compression $t_c$ and the time at maximum stylus velocity $t_v$. We define the dimensionless term $\tau = (t_c - t_v)/T$, where $T$ is the period of the engine oscillations. The two-spring model, CFD, and plate experiments show that the superpropulsion peak $\lambda_{max}$ occurs when the maximum compression of the projectile coincides with the maximum velocity of the engine at $\tau = 0$ (Fig. 2d). Interestingly, in the case of sharpshooters, the max compression time of the droplet also occurs around or before the maximum velocity of the stylus. These results suggest that the sharpshooters exploit the temporal tuning between the stylus and droplets for excretion (see Fig. 3 and next section).

To further understand the physics of superpropulsion, we consider the frequency ratio $f_o/f$ in light of the energy transfer between the stylus and the droplet[11]. The work $W$ done by the lower spring on the upper system from $t = 0$ until the upper mass reaches its maximum velocity at ejection $t = t_e$ is calculated as $W_{t_o \to t_e} = \int_0^{t_e} F_d(t)V_s(t)dt$, where $F_d = k_d(z_d - z_s)$ is the force on the upper mass and the $\dot{z}_s$ is the speed of the underlying stylus. We show that the normalized maximum energy transfer $\bar{W}_{max}$ occurs at $(f_o/f)^* \sim 3.23$ where $\lambda$ is maximum ($\lambda = 1.76$) and $\tau = 0$ (Fig. 2d). Thus, maximum energy transfer between the engine and the projectile results from maximizing the force and velocity product (power) over the ejection cycle, which is modulated by the temporal delay caused by the elasticity of the projectile[11,15]. In fact, the work done may be negative during parts of the cycle for $f_o/f < 1.16$ (Supplementary Fig. 8). Together, these results reinforce the significance of temporal tuning between the projectile and engine to achieve peak kinematic performance.

### Tight temporal tuning between droplets and stylus

Can we disrupt this tuning between the droplet and the stylus to yield sub-propulsive sharpshooters? We perform ablation experiments where we mechanically trim the tip of the hair structures (sensilla) located at the apex of the anal stylus of glassy-winged sharpshooters ($n = 5, N = 10$) (Fig. 3a, Supplementary Information, Section III). Two-tailed Mann-Whitney U test shows that 'hairless' sharpshooters eject smaller droplets having diameters of $D_o^- = 507 \pm 50 \, \mu m$ ($p < 0.02$) at higher speeds for their stylus $V_s^- = 0.83 \pm 0.16 \, m/s$ ($p < 0.02$) (Fig. 3b, Supplementary Information, Section III) with no significant change in

ejected droplet speed. Strikingly, the speed ratio $\lambda^-$ is <1 indicating that droplet ejection in hairless sharpshooters does not fall in the superpropulsion regime (Fig. 3b).

We analyze temporal tuning represented by the frequency ratio $f_o/f$ in light of peak kinematic performance predicted by the two-spring model. Hairless sharpshooters pump smaller droplets resulting in higher values of $f_o \propto D^{-3/2}$. Additionally, we observe that the frequency of the hairless stylus $f$ also increases (Supplementary Information, Section III). However, the resulting $f_o/f$ are not conserved and are lower than their control counterparts ($p < 0.02$, two-tailed Whitney-Mann U test) (Fig. 3b). The control sharpshooter species demonstrate a tightly tuned system—a characteristic of power amplified ultrafast systems[16]—as they lie within the $(f_o/f)^* \sim 3.23$ regime, corresponding to peak kinematic performance (Figs. 3d and 2c, d). In contrast, the stylus frequencies in the hairless case are scattered well above this window, clearly highlighting a disruption in the temporal matching between the stylus and the droplets. This mismatch is further substantiated by examining the droplet dynamics during droplet ejection, including the larger droplet deformation and rotational effects (Supplementary Information, section III).

### Parahydrophobicity of the anal stylus actuator

Next, we consider the effect of the surface properties of the stylus on the dynamics of droplet ejection. Detaching a droplet efficiently from a rotating substrate requires two competing demands: low surface energy for easy detachment and sufficient adhesion, so the droplet does not roll off prematurely. We observe that the stylus is parahydrophobic, having both a high apparent contact angle $\theta_a$ and strong surface adhesion, a motif shared by other natural (rose petals) and synthetic substrates[24,25]. This parahydrophobicity is evidenced by both the constant contact area $S$ during droplet growth and observations that the droplets do not roll off due to gravity or stylus orientation (Fig. 1d, Supplementary Information, Section IV). During emission of vibrational mating calls (while excreting), droplets can be subjected to violent displacements of up to~10 $\mu m$ and speeds ($V_d - V_s$) of ~2 $cm/s$ without detaching the droplet (Fig. 3, Supplementary Video I)[26]. We assume that droplet ejection occurs when the stylus kinetic energy $E_k \propto \frac{1}{2}\rho V - (2\pi f A)^2$ (where $A$ is the amplitude of the arclength set by the stylus tip and $V$ is the volume of the droplet) overcomes the surface adhesion $E_s \propto S\gamma(1 + cos\theta_a)$ (Supplementary Information, Section IV and Supplementary Fig. 6). Taking the ratio $E_k/E_s$ yields a critical frequency ratio $(f_o/f)_c$ required to detach a droplet $(f_o/f)_c \sim 4A\sqrt{\pi/(3S(1 + \cos\theta_a))} \leq 5.2 - 13.12$ for a $\theta_a = 100^\circ - 150^\circ$, respectively. The impact of surface adhesion on superpropulsion is further supported by numerical CFD simulations on droplets with different contact angles (Fig. 3f, Supplementary video II). These results suggest that superhydrophobicity and adhesion are important for stable droplet ejection in fluctuating environments. Additionally, these properties enable larger droplet volume for a given contact area $S$, resulting in storage of more elastic energy through surface deformations $dh \sim R^{3/2}V$[17] and lower critical velocities and accelerations[27-29]. A more accurate analysis of droplet detachment would include understanding the contact line's dynamics and the droplets' shape during the vibration cycle.[25] (Supplementary Information, Section IV and Supplementary Fig. 9)

### Energetics of superpropulsive excretion

We finally explore if superpropulsion could be energetically advantageous for these insects with a nutrient-limited food source. The net energy gain per unit volume (Energy Density) per feeding-excretion cycle for the organism can be estimated as:

$$\eta \geq (\eta_x - \eta_f) - \eta_{ex} \tag{1}$$

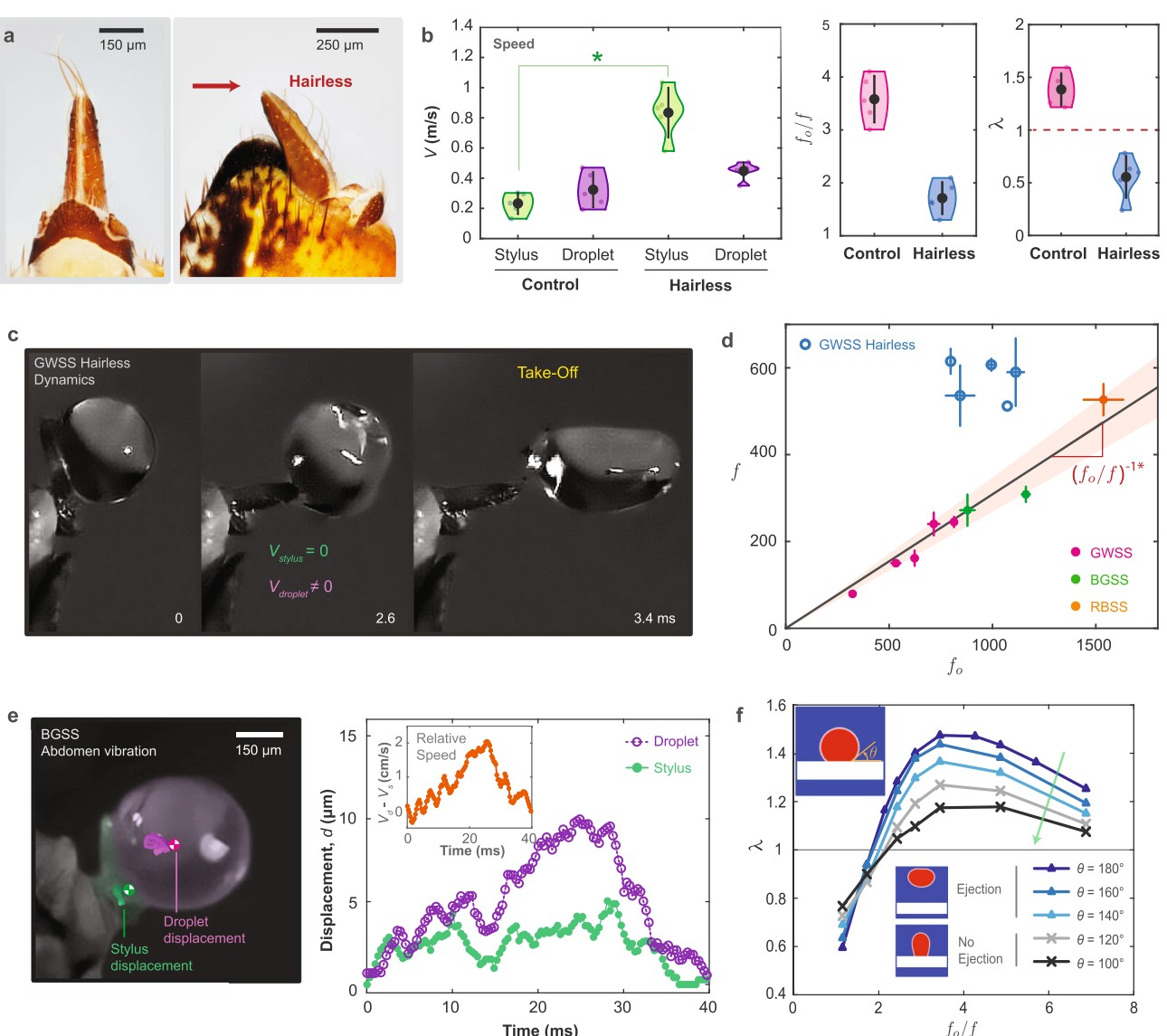

**Fig. 3 | Temporal tuning in superpropulsion and role of surface adhesion from droplet stability and ejection. a** We trim the mechanosensitive hair structures (sensilla) located at the top of the anal stylus to disrupt temporal tuning between the droplet and the stylus. **b** The maximum speed of the stylus ($V_s^-$) is higher than in control sharpshooters (two-tailed Whitney-Mann U test, $*p = 0.011$, $n = 5$ hairless individuals, $n = 5$ control individuals). In addition, the calculated frequency ratio $f_o/f$ is lower than the expected frequency ratio in control sharpshooters, whereas the speed ratio $\lambda < 1$ indicates a disruption in superpropulsion. **c** High-speed imaging shows that ejected droplets undergo significant deformation, and take-off occurs after the stylus stops moving. **d** The mean stylus frequencies $f$ in control sharpshooters lie within a tight window of $(f_o/f)^{-1} \sim 0.27 - 0.35$ (inverse of $f_o/f$), which falls around peak kinematics and energy transfer at droplet take-off. This matching is disrupted in hairless sharpshooters where associated $f$ are scattered away from that window. **e** The anal stylus exhibits strong capillary adhesion where excreted droplets remain adhered despite lateral and vertical displacements $V_d - V_s \sim 2\ cm/s$ observed during mating calls. **f** Computational fluid dynamics (CFD) simulations reveal a theoretical limit for droplet ejection of a sessile droplet having a contact angle $\theta$ (Supplementary Video II). Sessile droplets with a relatively high contact angle ($\theta > 130$) do not take off from the surface of the vibrating plate. Superpropulsion is conserved where $\lambda = V_d/V_s > 1$ even if ejection does not occur. Due to adhesion, the maximum speed of the droplet is not equivalent to its ejection speed. Error bars of all data points in this figure represent an average value ± one standard deviation.

where $\eta_x$ is the energy content of nutrients in xylem sap, $\eta_f$ is the energy per unit volume expended by the cibarial pump during feeding, and $\eta_{ex}$ is estimated from the energy required during excretion. Maintaining net positive energy gain ($\eta$) from feeding on xylem fluid is critical for the insects' functioning and survival[5,30]. Andersen et al.[5] estimated the net energy of xylem sap and cibarial pumping at negative pressures for sharpshooters, i.e. $\eta_{in} = (\eta_x - \eta_f)$ as $2 \times 10^5 - 8.2 \times 10^6\ J/m^3$ while considering factors such as xylem tension, time of the day, metabolic cost, the efficiency of the cibarial pump and nutrient concentration in xylem fluid. This range sets the upper limit for $\eta_{ex}$ since any higher energy expenditure on excretion will place the organism in an energy deficit.

To estimate $\eta_{ex}$, we model the excretion of fluid through the anal tube as a pressure-driven flow of liquid water through a straight and circular cylinder having a diameter $d$ and length $l$ estimated from microCT (Fig. 4b, Supplementary Information, Section IV, Supplementary Video III and Supplementary Fig. 1). We estimate the average pressure (since $1\ Pa = 1\ J/m^3$) using the energy balance equation to pump water at a steady state with a flow speed $u$ across the hindgut and form a droplet on the stylus as following[31] (Supplementary Information, Section V):

$$\eta_{ex} = \frac{32\mu l u}{d^2} + \frac{4\gamma}{d}, \qquad (2)$$

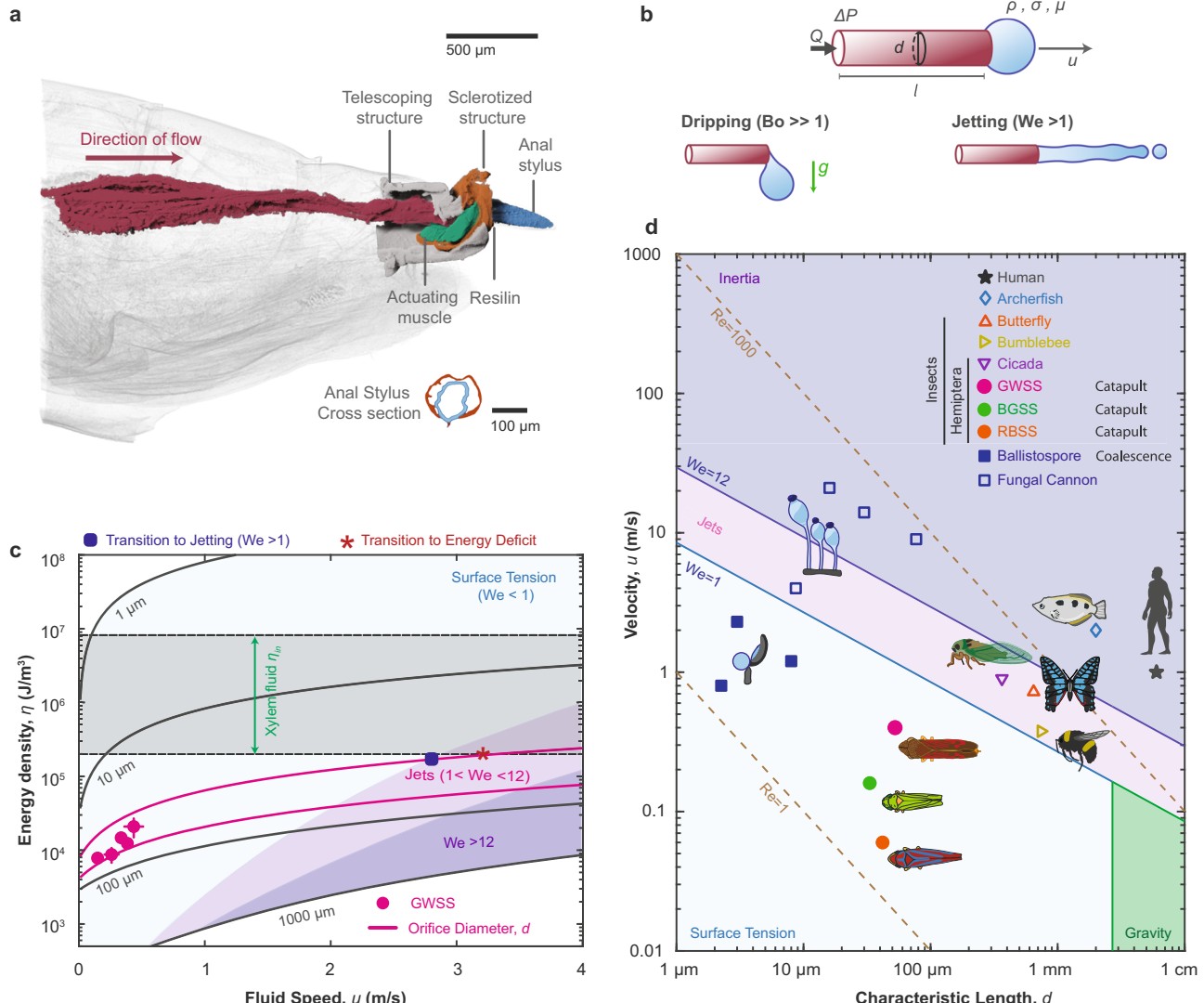

**Fig. 4 | Energetics of superpropulsion and physical scaling. a** MicroCT scan of a GWSS revealing the morphology of their hindgut. Digested fluidic waste is excreted through the rectum into the narrow anal stylus before being ejected. **b** The hydrodynamics of excretion in sharpshooters is modeled as a pressure-driven flow across a cylindrical tube. Fluid exiting a nozzle forms a pendant droplet at low speeds ($Bo \gg 1$ and $We < 1$) or jet when inertial forces are dominant ($We > 1$). **c** The pressure (Energy per unit fluid volume) exerted by sharpshooter insects increases proportionally to the speed of the exiting fluid and the orifice diameter. At their small scale, forming jets to expel their fluid waste is 4 × more energetically taxing than forming individual droplets and flicking them **d** Small organisms living in a surface tension-dominated world face the challenge of overcoming capillary adhesion. Ballistospores and sharpshooters exploit surface-tension properties (coalescence and superpropulsion) to eject droplets at high speeds. Error bars represent average value ± one standard deviation.

From eqn. (2), we see that the energetic cost of pumping fluid depends on the flow's speed and the tube's size. The average energy per unit volume to form a droplet for a sharpshooter insect (GWSS) $\eta_{ex} = 1.3 \times 10^4 \, J/m^3$ ($N = 22$), which is lower than $\eta_{in}$ ensuring a net positive energy gain to meet other biological functionalities (Fig 4c). However, transitioning to a 'jetting' regime instead of droplet-formation requires increasing the speed of the exiting flow, which is energetically costly ($\eta_{ex} \propto u$, Fig. 4c). Specifically, for GWSS, we estimate that ejecting a droplet requires 4–8 × less energy than forming a jet at $We \geq 1$, i.e., when inertial forces overcome surface tension forces[32,33](Fig. 4c, Supplementary Information, Section V).

Once a droplet is formed on the stylus, to overcome surface adhesion, kinetic energy is imparted by the stylus to eject the droplet (either sub- or super-propulsively). The average kinetic energy per unit fluid volume $\bar{E}_k \sim \frac{1}{2}\rho V_d^2$ is $\sim 10^2 \, J/m^3$, which is two orders of magnitude less and negligible compared to $\eta_{ex}$. Furthermore, we also find that the energy required to eject droplets in superpropulsion mode is lower

compared to subpropulsion when $f_o/f < (f_o/f)_c$, suggesting energetic penalties associated with subpropulsion as $\bar{E}_k \propto f^2$ (Supplementary Information, Section II). These results demonstrate the ultra-low energy cost of the superpropulsion mechanism in sharpshooters.

## Fluidic ejections across physical scales

Lastly, we discuss how the fluidic excretion mechanism of sharpshooters compares with the other organisms excreting fluid through an orifice (Fig. 4d). The methods used by organisms for such ejections are diverse and influenced mainly by the organism's behavior and dominant forces at its length scale. These forces are typically characterized by the Reynolds number ($Re$), the Bond number ($Bo$), and the Weber number ($We$) (Fig. 4d). Animals having an orifice size $d > 100 \, \mu m$ (mammals, archerfish, and large insects) rely on inertial forces for repeatable fluidic ejection by forming jets or large sheets of liquids during urination[31] or hunting[34]. Smaller organisms, however, living in a surface-tension-dominated world

($Bo \ll 1$ and $We < 1$) develop ingenious mechanisms to overcome capillary adhesion.

Sharpshooters (studied here) use their specialized resilin-powered and tuned anal stylus to transfer kinetic energy to the droplets. A similar droplet catapulting strategy is also observed in phloem fluid-feeding aphids (Hemiptera: Aphididae) that kick their sugar-rich excrement (with either their legs or caudas). Aphids also manipulate surface tension forces by coating their excrement with powdery wax, transforming them into a non-sticky liquid marble[35]. Not all xylem-feeders employ droplet ejection to eliminate their fluidic waste. Cicadas (Hemiptera: Cicadidae) form powerful jets to remove their excreta because they are ~2–7 × larger than an adult sharpshooter (Fig. 4d)[36] and do not share the same energetic constraints faced by the sharpshooters. Outside the animal kingdom, tiny fungal species such as ballistospores and fungal cannons with characteristic length $1 \mu m < d < 100 \mu m$ resort to explosive jets or droplet coalescence to disperse their spores[37]. Both these fluidic ejections mechanisms are destructive and non-repeatable, unlike the one used by the sharpshooters. Collectively, these diverse fluid ejection techniques highlight the importance of the physical and energetic constraints in governing the mechanism in a given organism.

## Discussion

In summary, our work reveals that droplet ejection through superpropulsion serves as a strategy to conserve energy per feeding-excretion cycle for these tiny, xylem-sap feeders. Such an energy-efficient strategy for excretion is necessary because the increased miniaturization of these xylem-feeding insects (e.g., juveniles or smaller species) leads to higher energetic losses during fluidic pumping due to viscous and surface tension forces $\eta_{ex} \propto d^{-2}$, suggesting a fundamental limit on the minimum body size of xylem-sap feeders[38]. In addition to energetic advantages, flinging droplets to large distances could reduce the chances of detection by predators such as the parasitic wasp *Cosmocomoidea*, which may be attracted to chemical cues in the accumulation of sharpshooters excreta[39] (Supplementary Video I).

The need to create a considerable distance between insects and their waste is mainly observed in shelter-dwelling or site-faithful insects, which are typically pressured to maintain hygiene at the site to avoid the growth of pathogens and reduce chemical cues for potential predators. The ballistic ejection of excrement is not uncommon among insects. Many insect species may be described as 'frass-shooters,' 'butt-flickers,' and 'turd-hurlers' that innovated unusual strategies to launch away both liquid and solid excrements[2]. For instance, frass-shooting skipper larvae employ biological latches on their anal plates coupled with a hydrostatic blood pressure buildup to propel solid pellets up to 38 times their body length with a speed larger than 1.5 m/s. Some species of noctuids violently shake their abdomen as they release their frass pellets, and some geometrid larvae use their thoracic legs to kick away their frass pellets[2]. Insects, in particular, exhibit a wide range of ingenious waste disposal strategies determined by their environment, diet, and generalized lifestyle. Thus, by conducting a detailed analysis of excretion dynamics in the sharpshooter insect, our work reinforces the importance of studying waste elimination to reveal a holistic perspective of organismal behavior.

Finally, our study of the droplet catapulting mechanism in sharpshooters offers a first observation and quantification of superpropulsion in a living organism. At its core, superpropulsion provides a gateway to propel an elastic projectile faster than the maximum speed of its actuator through temporal tuning and can be viewed as a single-shot resonance system[10]. For physical systems dominated by surface tension forces, superpropulsion offers an ingenious mechanism through which this impeding force can be harnessed as a spring, albeit with tight temporal tuning for a full advantage. However, this superpropulsion mechanism may not be unique to surface tension-dominated systems. It may be exploited by other small power amplified and impulsive biological systems that rely on elastic structures to overcome the power limits of striated muscles[16]. Our work takes these insights from nature and provides a fundamental framework to implement an energy-efficient superpropulsion mechanism for manipulating elastic objects in synthetic systems ranging from pick and place nano- and micro-fluidic devices to smart wearable electronics and soft, elastic robotic engines[10,11,15,40].

## Methods

### Two-spring model

We describe the dynamics of the stylus and the droplet with a minimalist mathematical model based on a coupled double mass-spring-damper inspired by the wave model[10], chained spring systems[11,15] and bouncing model[28] (Fig. S5). The upper mass-spring-damper represents the water droplet, and the lower mass and spring represent the resilin compression at the stylus. We ignore dissipation in the resilin-based spring due to the high efficiency of resilin[19].

Elastic deformation in water droplets is mediated by surface tension. We consider the Weber number $We = \rho R_o V_d^2/\gamma$, which relates the relative effect of the inertial forces from the forceful impact relative to surface tension. For a water droplet (Surface tension: $\gamma = 72 \, mN/m$, Density: $\rho_w = 996 \, kg/m^3$), the Weber number is $We \sim 10^{-1}$ indicating the dominance of the surface forces. Also, since the acceleration of the actuating substrate is relatively low (14–28 $g$, where $g$ is the gravitational constant), the droplet remains in the linear droplet deformation regime and does not form water puddles[14]. Under these conditions, water droplets ($Bo < 1$) deform from a spherical cap into an oblate spheroid and behave as a quasi-Hookean spring with a stiffness $k_d = 32\pi\gamma/3 = 2.41 \, N/m$[17,18].

Resilin is a super elastic protein known for its high efficiency (~97% efficiency) in loading and releasing energy in insect movement[19]. Natural resilin has an elastic modulus ($E_r$) ranging between 0.1–3 $MPa$[41]. During the spring loading phase, the stylus bends and deforms the resilin blob into a complex shape. To estimate the effective spring constant $k_s$, the tiny resilin blob is assumed to obey Hooke's law of elasticity throughout compression and extension cycles. For simplicity, we also assume that the resilin blob is a sphere with an area of contact with the stylus $S_r \sim 0.25$–$1 \times 10^{-8} \, m^2$ estimated from video analysis and close-up microscopic imaging. Based on these simplifying assumptions, the effective spring constant of the resilin is calculated using $k_s = S_r E_r/d \sim 2.5$–600 $N/m$, where $d$ is the maximum compression of that blob, assumed to range ~50–$100 \times 10^{-6} \, m$. We note that the ratio of $k_d/k_s \sim 0.004$–1 indicates little to no coupling between the two springs. In addition, the effective mass of the stylus combines the droplet and the stylus $m_{eff} = m_d + m_s > m_s$ indicating no inertial coupling.

We write the 1D coupled equations of motion in the upward $z$-direction as follows:

$$M\ddot{z} + C\dot{z} + Kz = 0$$

$$\begin{bmatrix} m_s & 0 \\ 0 & m_d \end{bmatrix} \begin{pmatrix} \ddot{z}_s \\ \ddot{z}_d \end{pmatrix} + \begin{bmatrix} c_d & -c_d \\ -c_d & c_d \end{bmatrix} \begin{pmatrix} \dot{z}_s \\ \dot{z}_d \end{pmatrix} + \begin{bmatrix} k_s + k_d & -k_d \\ -k_d & k_d \end{bmatrix} \begin{pmatrix} z_s \\ z_d \end{pmatrix} = 0$$

(3)

The simulations start by compressing the lower spring with a displacement $dz_s$ below the equilibrium point (replicating the deformation of resilin during the spring loading phase). At $t = 0$, the compressed lower spring is released. The spatiotemporal dynamics of the two springs are obtained by numerically solving the system of equations of motion using the 4th-order Runga-Kutta method in MATLAB.

Dissipation within sessile droplets is modeled as a simple linear damper. Origin of damping includes bulk viscous dissipation, interfacial boundary layer damping, and moving contact line[20–22]. The effect of dissipation is gauged by running simulations for different values of damping ratio $\zeta = c/2\sqrt{k_d m_d}$ ranging from $\zeta = 0$ (no dissipation) to $\zeta = 1$ (critically damped) (Fig. S5). We observe that as damping increases, the effect of elastic advantage decrease where the elastic projectile gradually behaves like a rigid projectile ($\lambda \to 1$ as $\zeta \to 1, \forall f_o/f$). In addition, increasing the damping coefficient $\zeta$ shifts $\tau$ since the frequency of the upper spring now corresponds to the damped $f_d = f_o\sqrt{(1 - \zeta^2)}$ (Fig. S5).

## Computational fluid dynamics (CFD)

Using time-dependent two-phase flow physics, the finite element simulations of a 2D droplet on a moving plate are performed in COMSOL Multiphysics 5.6. Navier-Stokes equations are solved in the liquid and air domain. Liquid/air/solid interfaces are modeled using the level-set method. Before prescribing the motion to the plate, the droplet is allowed to equilibrate on the stationary plate to achieve its equilibrium shape based on the prescribed contact angle. Following equilibration, the plate is prescribed a sinusoidal displacement using a moving mesh interface, where it is modeled as a moving boundary within a deforming domain. Automatic remeshing of the domain is performed at specified timesteps to prevent excessive deformation of the mesh elements because of the moving boundary. For the outer boundaries of the domain, we prescribed the outlet boundary condition, specifying the static pressure to zero. For the wetted wall, i.e., the surface of the plate, we defined a static contact angle with a Navier-slip equal to the maximum size of the mesh element. We used a free triangular mesh with a maximum element size of 0.015 mm and a minimum of 0.0001 mm. The reinitialization parameter for the level set method is set to 0.5 m/s, and the parameter controlling interface thickness is equal to the mesh element's maximum size. We simulated droplets at different contact angles ranging from 100°–180°. For 2D simulations, we fixed the area of the droplet corresponding to the droplet's diameter of 1 mm, forming a perfect circle and using this same area to equilibrate droplets under gravity at different contact angles. Given no coupling between the upper and lower springs (Supplementary Information, Section II, B), we prescribed the sinusoidal motion to the plate of the form $z = A\sin(2\pi f_s t - \pi/2)$ where $f$ is the frequency of plate at a constant peak acceleration of 140 m/s$^2$ with the frequencies of vibration ranging from 50 Hz to 300 Hz. We have also performed simulations for droplet sizes 1.2 mm and 1.4 mm, and for acceleration 80 m/s$^2$ which show good match with the drop-on-plate experiments (Fig. 2 and Supplementary Fig. 10). However, we didn't find significant differences between the superpropulsion curves when compared with a droplet of 1 mm size.

## Experiment: Drop-on-plate

We analyze the ejection of 1 mm droplets sitting on a vertically vibrating plate with a superhydrophobic surface $\theta_e \sim 180°$. A sinusoidal displacement $z = A\sin(2\pi f_s t - \pi/2)$ is prescribed to the plate with $A$ as the amplitude of displacement, $f_s$ is the frequency of vibration, and $-\pi/2$ is the phase shift to replicate the compression of the lower spring at $t_0 = 0$. Similar to the two spring models, simulations are repeated for various $f_o/f_s$ by changing the input vibrational frequency $f_s$ while keeping $f_o$ constant (same droplet size). The experimental data capture the overall shape of the theoretical and computational curves over a wider range $f_o/f_s$.

## Reporting summary

Further information on research design is available in the Nature Portfolio Reporting Summary linked to this article.

## Data availability

The raw data that support the findings of this study are available in https://github.com/bhamla-lab/Sharpshooter_Nat_Comm_2023 and https://doi.org/10.5281/zenodo.7537826.

## Code availability

The code used to generate the data for the theoretical model are available in https://github.com/bhamla-lab/Sharpshooter_Nat_Comm_2023 and https://doi.org/10.5281/zenodo.7537826.

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

## Acknowledgements
We thank Sheila Patek, Mark Ilton, Christophe Raufaste, Christopher H. Dietrich, Elizabeth Clark, Pankaj Rohilla, Jacob Harrison, Ishant Tiwari, Jonathan O'Neil, Rodrigo Almeida, and Elaine Backus for their valuable help, feedback, and suggestions. Thanks to Andrea Clark (Micro and Nano CT Imaging, University of Michigan) for the MicroCT scans. Thanks to Soham Sinha, Raghav Acharya, and Shuvam Samadder for their initial help with data collection, data analysis, and computational modeling, and to the rest of the Bhamla Lab for the helpful discussions and comments. Funding acknowledgments: NIH MIRA Grant R35GM142588 (M.S.B.); NSF Grant MCB-1817334 (M.S.B.); NSF CAREER IOS-1941933 (M.S.B.); the Open Philanthropy Project (M.S.B.).

## Author contributions
Conceptualization: E.J.C., M.S.B. Methodology: E.J.C., P.S., R.K., M.S.B. Data Analysis: E.J.C., P.S., M.S.B. Investigation: E.J.C., P.S., R.K., Supervision: M.S.B. Writing—original draft: E.J.C. Writing—review and editing: E.J.C., P.S., R.K., M.S.B.

## Competing interests
The authors declare no competing interests.

## Additional information

**Peer review information** . *Nature Communications* thanks Zhizhao Che, Thierry Darmanin and the other, anonymous, reviewer for their contribution to the peer review of this work. Peer reviewer reports are available.

