## [Peer Review File · Nature Communications]

REVIEWER COMMENTS

Reviewer #1 (Remarks to the Author):

The manuscript reports a droplet superpropulsion phenomenon that sharpshooters use for excretion. In this superpropulsion process, sharpshooters can catapult the droplet with a droplet speed faster than their own movement speed. To explain this process, a two-spring model is proposed, and numerical simulations of droplets on vibrating surfaces are performed as well. The authors find the superpropulsion is achieved by the temporal turning between the droplet and the stylus. The authors also explained the role of surface parahydrophobicity on the superpropulsion phenomenon, and discussed the fluid ejection modes used by organisms of different scales. The results presented in this study are novel and interesting. However, there are several major and minor issues in this manuscript as follows.

1. The spring model is often used to describe the oscillation with a sufficiently long time, i.e., the equilibrium state. However, for the problem considered in this study, the stylus motion is a single kick, and the system is far from equilibrium. Therefore, the process should highly depend on the mode of actuation. For example, if you oscillate a spring system far from its natural frequency, the motion of the mass will be very complex. However, this study simply considers the natural frequency. Therefore, the validity of the spring model should be further checked.
2. In the propulsion process, the motion of the stylus is basically rotational (the authors use angular speed and angular acceleration to describe it). However, in the spring model, the motion of the stylus is simply translational. Can the translational movement in the model capture the rotational movement of the stylus? Does the rotation of the droplet affect the translation?
3. In the CFD simulation, the authors consider the take-off of a droplet on an oscillating substrate, which is often considered as a spring. However, in the model, there are two springs, the droplet and the stylus. How did the authors consider the spring of the stylus? If only the droplet is simulated, how did the authors obtain the values for the stylus, such as f and λ ? It is suggested that the authors include the motion and the deformation of the stylus into the simulation.
4. The simulation is two dimensional. A 2D droplet corresponds to a cylindrical shape, and has a different oscillation frequency.
5. In the analysis of the critical condition for droplet detachment, a key assumption of the energy conservation model is that the kinetic energy of the droplet exceeds the surface adhesion energy. It is fine to use energy balance to determine the critical condition, but droplet detachment is a local phenomenon, i.e., the surface of the droplet is completely detached from the surface of the stylus. A detachment will occur as long as there is still upward local motion around the detachment area when the surface adhesion energy approaches zero. Because there is strong deformation of the droplet, the velocity within the droplet will be different. Hence, the authors should justify this assumption, for example, by considering how the local phenomenon is connected to the overall energy budget.

6. The caption of Fig. 2d is missing.
7. What is Fig. 1f? The symbol is θ , but the unit is $^\circ/\text{s}$, i.e., for angular velocity.
8. How is $t=0$ defined? How are the max compression time t_c and ejection time t_e defined? Do they mean a period or a moment?
9. There are many typos in the manuscript. Just to name a few:
 - a. In the heading on Page 5, 'Parahydrophibicity' should be 'Parahydrophobicity'
 - b. In the caption of Fig. 3, 'then' should be 'than'.
 - c. In the caption of Fig. 5 in Supplementary Material, 'te' should be ' t_e '.

Reviewer #2 (Remarks to the Author):

Report on the manuscript « Droplet superpropulsion in an energetically constrained insect »

As stated by the authors, this paper « present the superpropulsion phenomenon for the first time in a biological system » and in that sense it is really interesting and suitable for publication. The paper is well written and several (maybe too much ...) points are addressed (experimental, theoretical and simulations). Nevertheless, following suggestions have to be considered/answered before publication.

1) A much more deeper presentation of the super propulsion phenomenon has to be addressed.

The sentence « Conceptually, this is similar to an Olympic diver perfectly timing their launch off a springboard. » is not appropriate at all ; in that case the energy come from the projectile and not from the catapult. Such a short-cut has to be removed and I recommend a much more detailed discussion of the initial papers (ref 10 and 11). References to :

* Giombini et al. (Phys Rev E 2022) (spring model, effect of substrate elasticity)

* Chantelot et al (Phys. Rev Fluids 2018) (large accelerations)

* Celestini et al. (J. Coll Int. Sci 2018) (effect of surface tension and viscosity)

have to be added to clearly illustrate the discussion/presentation of the superpropulsion phenomenon.

2) The two spring model has already been proposed (ref 11 and one of the ref proposed just above) and I think it is better to much more concentrate on CFD simulations that present new results.

3) Ejection experiments of drops standing on a plate have also already been performed in the initial paper presenting superpropulsion (ref 10). I wonder if it is necessary to present these results in this paper. The discussion proposed in point 1) just above will do the job.

4) Figure 2 : the fig 2-d is not commented in the caption. Remove the label d) in the figure.

5) The restitution coefficient is defined in terms of velocity. Why the authors did not present it in terms of energy as initially done in ref 10. It will help to compare both experimental and theoretical results.

6) Have you measure the surface tension of ejected droplets ? In this paper you actually suppose it is water ... Is this true for excretion of GWSS ?

Reviewer #3 (Remarks to the Author):

In this original work, the authors report, for the first time, the superpropulsion phenomenon in a biological system. Their study was extremely well performed by studying for example the shape of the droplet or the elasticity of the projectile. I have just some slight comments for this manuscript:

-The term "Parahydrophibicity" should be changed by "Parahydrophobicity". For the parahydrophobicity, the authors should also cite and use this recent work: Experimental characterization of droplet adhesion: the ejection test method (ETM) applied to surfaces with various hydrophobicity, J. Phys. Chem. A, 2018, 122, 8693–8700.

-The authors say "a motif shared by other self-cleaning natural (rose petals) and synthetic substrates." Rose petals do not have self-cleaning properties.

Responses to comments from Reviewer #1

We would like to thank Reviewer #1 for the time and effort required for the detailed comments, especially about the validity of the two-spring model and the importance of droplet detachment. We have tried our best to address most of the comments, some of which we will try to incorporate into our ongoing work. The insights and suggestions provided in the review document helped us improve the manuscript.

Please find below our point-by-point response. We include two versions of the revised manuscript: an annotated version highlighting all the changes/additions and a non-annotated version. In the revised manuscript, we include a new methods section where we describe the two-spring model and Computational fluid dynamics model, two additional figures in the supplementary information (Fig. S9 and Fig. S10), a new paragraph in the superpropulsion of liquid droplets section, and several minor modifications and additions spread throughout the manuscript.

In the rebuttal document below, the referees' comments are in black, and the author's responses are in blue.

The manuscript reports a droplet superpropulsion phenomenon that sharpshooters use for excretion. In this superpropulsion process, sharpshooters can catapult the droplet with a droplet speed faster than their own movement speed. To explain this process, a two-spring model is proposed, and numerical simulations of droplets on vibrating surfaces are performed as well. The authors find the superpropulsion is achieved by the temporal turning between the droplet and the stylus. The authors also explained the role of surface parahydrophobicity on the superpropulsion phenomenon, and discussed the fluid ejection modes used by organisms of different scales. The results presented in this study are novel and interesting. However, there are several major and minor issues in this manuscript as follows. .

1. The spring model is often used to describe the oscillation with a sufficiently long time, i.e., the equilibrium state. However, for the problem considered in this study, the stylus motion is a single kick, and the system is far from equilibrium. Therefore, the process should highly depend on the mode of actuation. For example, if you oscillate a spring system far from its natural frequency, the motion of the mass will be very complex. However, this study simply considers the natural frequency. Therefore, the validity of the spring model should be further checked.

Spring models may be used to model transient (initial) movement and equilibrium dynamics across a wide range of forcing frequencies. The rationale behind this approach here is to highlight the importance of elasticity through temporal tuning in both the engine and the projectile (i.e. ω/ω_0) on the kinematic outcome of ejection and connected to the original work by **Ref. 10 (Raufaste *et al.*)**, **Ref. 11 (Celestini *et al.*)**, and **Ref. 15 (Giombini *et al.*)**. The solution of the equations of motion does not reveal any nonlinearities in the asymptotic cases for ω/ω_0 .

The similarity between the droplet-stylus/resilin and the double mass-spring is only assumed to be valid until ejection - beyond that point, complex dynamics due to the dynamic coupling between the upper and lower systems will occur, and the two-spring model is no longer valid. In fact, the model predicts a lower kinematic performance of the ejected projectile potentially caused by complex dynamics at higher/lower f_0/f .

We are attaching here several examples in the literature where spring models have been used to examine impulsive, single kick movements (where the timescale of actuation is essentially much smaller/higher than the natural frequency of the mass-spring) in the context of ejection of droplets and soft solids, throwing, and jumping/hopping which we reference in the main text:

Ref. 15: Giombino et al., 2022, PRE (<https://doi.org/10.1103/PhysRevE.105.025001>) and **Ref. 11: Celestini et al., 2022, PRA** (<https://doi.org/10.1103/PhysRevApplied.14.044026>)

These two papers employ a spring model to study throwing a compliant projectile using a similar model (two-spring model). Fig. 1 (in PRE) shows a wide range of frequency ratios from 0 to 10.

Ref. 18: Pierre Chantelot et al., 2018 EPL (<https://doi.org/10.1209/0295-5075/124/24003>)

This paper uses a two-spring system to model the impact dynamics and vibration of a water droplet on a soft substrate and models it as a two-spring system.

Ref. 28: Hubert et al., 2014, Physica D (<http://dx.doi.org/10.1016/j.physd.2014.01.002>)

This paper modeled the deformation dynamics of a droplet with two-mass connected by a spring following successive impacts on a vibrating bath.

Blickhan et al., 1989, J. Biomech. ([https://doi.org/10.1016/0021-9290\(89\)90224-8](https://doi.org/10.1016/0021-9290(89)90224-8))

This paper modeled running and hopping as a mass-spring system.

We included a method section in the main manuscript that describes the two-spring model in detail and a section in the supporting information that describes our model's underlying assumptions and limitations, which we include here for convenience:

"The two-spring oscillator model is chosen for its simplicity in describing the droplet's vibrations and resilin's dynamics. It is not meant to give a one-to-one representation of the system but to reveal critical features of two coupled oscillating systems, such as temporal matching and ejection speed. The complex dynamics of bouncing droplets have been previously modeled using different spring-based models, such as the 'bouncing model' whereby a droplet is estimated as a Kelvin-Voigt material [8]. A molecular dynamics approach is taken to evaluate the contact mechanics between the actuator and the droplet. The contact force is modeled as a damped spring only active during the compression of the upper spring and null otherwise. Similar to the two spring models, the bouncing model predicts superpropulsion within a different frequency range and λ vs. f_0/f profiles (Fig. S7).

Another simplification in this model is that we ignore any potential ejection dynamics arising due to the sessile nature of the formed water droplet. Sharp et al. showed that the resonant frequency response of sessile droplets depends on the contact angle such $f_i = (\pi/2)\sqrt{(i^3\gamma \cos^3 \theta_e - 3 \cos \theta_e + 2)/(24m_d\theta_e^3)}$ where i represent the i th vibrational mode of the droplet, and θ_e is the contact angle between and m_d is the mass of the quasi-spherical droplet [9]. This equation has a similar functional form to the classical equation developed by Lord Rayleigh for the natural frequency of free oscillating droplets (which is used in this work for $n = 2$), $f_n = 1/2\pi \sqrt{n(n-1)(n+2)\gamma/3\pi m_d}$ where $n = 2, 3, 4...$ corresponds to the mode number [9]. Adjusting the natural frequency f_o by considering the contact angle θ_e would shift the theoretical, computational, and field data, but the same trends are still conserved. "

2. In the propulsion process, the motion of the stylus is basically rotational (the authors use angular speed and angular acceleration to describe it). However, in the spring model, the motion of the stylus is simply translational. Can the translational movement in the model capture the rotational movement of the stylus? Does the rotation of the droplet affect the translation?

The reviewer makes a good point regarding rotation's effect on the droplet's ejection dynamics. In **supplementary information section II, C**, we examine the effect of rotation in the stylus and droplet on the translational kinematics of ejection by considering how much is transferred into rotation vs translation in the ejected droplets. Calculating the ratio of translational energy E_t to the rotational energy E_r of the droplet reveals that $(E_r/E_t) \sim 10^{-2}$. However, the rotational effects become more significant in the case of hairless sharpshooters, where the angular velocity increases from ~ 170 rad/s in the wild-type sharpshooter to $\sim 2.4 \times 10^3$ rad/s in the hairless case and the ratio $(E_r/E_t) \sim 0.2$. In this case, the two-spring model becomes invalid, which is the main reason for not including the hairless sharpshooters' data in the superpropulsion plots in the two-spring model and CFD. We discuss this in more detail in the revised **Supplementary information (Table 1 and section II, page 3)** and included below:

"Moreover, the ejected droplets undergo a substantial rotation rate around their axis with an angular velocity of $\Omega^- \sim 2.4 \times 10^3$ rad/s compared to $\Omega^+ \sim 170$ rad/s (Supplementary video I). To gauge the relative impact of rotation, we consider both the translational energy E_t and rotational energy E_r transferred to the droplet from the rotating stylus. In hairless sharpshooters, the energy ratio of ejected droplets $(E_r/E_t)^- = (rg\Omega/Vd)^2$ is ~ 0.2 (where $r_g = R_d/10$ is the radius of gyration of the spherical droplet). In control sharpshooters, however, energy is almost entirely transferred to translational energy with $(E_r/E_t)^+ \sim 10^{-2}$. This substantial increase in energy lost to droplet rotation implies that the rotational aspect of the stylus (previously ignored) plays an important role in dictating droplet dynamics during ejection. For instance, the increase in the frequency of the stylus f_s may lead to a considerable increase in the effect of inertial forces such as the Euler and centrifugal force ($\propto f^2$) in the rotating frame of reference of the stylus. Extreme cases of these inertial forces may lead to movement of the droplet (slip) in the parallel direction to the stylus, only to be opposed by capillary adhesion. As a result, the moving stylus may cause a net tangential force causing the droplet to rotate, similar to how an off-centered strike to

a billiard ball would make it sidespin. Furthermore, given the significant size of the hairs (at least 40% the length of the stylus), cutting the hair structures shortens the length of the stylus, possibly exacerbating these inertial and geometrical effects.”

Similarly, studies examined the physics of projectiles being launched from a rotating actuator **Ref. 12 (C. Cohen and C. Clarinet, <http://dx.doi.org/10.1051/ejn/2016301>)** while only considering the translational kinematics of the actuator and projectile. In the main manuscript, we include the following text in the to reference the above information in the Supplementary information:

“This mismatch is further substantiated by examining the droplet dynamics during droplet ejection, including the larger droplet deformation and rotational effects (Supplementary Information, section III)”

3. In the CFD simulation, the authors consider the take-off of a droplet on an oscillating substrate, which is often considered as a spring. However, in the model, there are two springs, the droplet and the stylus. How did the authors consider the spring of the stylus? If only the droplet is simulated, how did the authors obtain the values for the stylus, such as f and λ ? It is suggested that the authors include the motion and the deformation of the stylus into the simulation.

The elasticity of the droplet is mediated by surface tension, while the source elasticity of the ‘stylus (substrate)’ is the elastic resilin blob we discovered at the bottom of the stylus (Main text, Fig. 1b, and Supplementary information, Fig. 1). As discussed in **supporting information, section III, B- Resilin Spring**, given that the ratio of the effective spring constants of the droplet and resilin $k_d/k_r \ll 1$ and $m_d/m_{\text{eff}} < 1$, we assume that there is no coupling between the upper and lower springs. For this reason, to reduce the complexity of the simulations, the lower spring was replaced with a prescribed forcing sinusoidal function $z_s = A \sin(f_r - \pi/2)$ where f is the stylus frequency (actuating substrate) and the $\dot{z} = Af$.

Furthermore, the stylus (substrate) is assumed to be rigid since we do not observe any detectable deformation in high-speed videos. This is substantiated by the confocal and imaging which reveal that a stiff and sclerotized chitinous material covers the stylus.

We included this in the methods section describing the two-spring model, and we highlight this section below:

“Resilin is a super elastic protein known for its high efficiency (~ 97% efficiency) in loading and releasing energy in insect movement [10]. Natural resilin has an elastic modulus (E_r) ranging between 0.1–3 MPa [11]. During the spring loading phase, the stylus bends and deforms the resilin blob into a complex shape. To estimate the effective spring constant k_s , the tiny resilin blob is assumed to obey Hooke’s law of elasticity throughout compression and extension cycles. For simplicity, we also assume that the resilin blob is a sphere with an area of contact with the stylus $S_r \sim 0.25 - 1 \times 10^{-8} \text{ m}^2$ estimated from video analysis and close-up microscopic imaging. Based on these simplifying assumptions, the effective spring constant of resilin is calculated

using $k_s = S_r E_r / d \sim 2.5 - 600 \text{ N/m}$ where d is the maximum compression of that blob assumed to range $\sim 50 - 100 \times 10^{-6} \text{ m}$. We note that the ratio of $k_d/k_s \sim 0.004 - 1$ indicates little to no coupling between the two springs. In addition, the effective mass of the stylus combines the droplet and the stylus $m_{\text{eff}} = m_d + m_s > m_s$, indicating no inertial coupling.”

We added a section describing the computational fluid dynamics in the methods section describing these assumptions (highlighted in the text). We also adjusted the caption of the supplementary figure Fig. S3 to remove any potential misunderstanding.

We hope that the revised manuscript clarifies these details.

4. The simulation is two dimensional. A 2D droplet corresponds to a cylindrical shape, and has a different oscillation frequency.

While we agree that a 3D geometry of the droplet would more accurately capture the vibrational frequency of the droplet, 3D simulations would significantly increase the complexity and runtime of the simulations without adding significant support to the main conclusions of this paper.

The main purpose of the CFD simulations is to complement the theoretical model, organismal data, and experimental data on establishing the principles of superpropulsion and showcase how temporal tuning influences the kinematics of droplet ejection. Building on this model, we were able to start exploring surface effects (such as adhesion) on the outcome of superpropulsion. The CFD model does capture the superpropulsion trend and matches well with the experimental data (drop-on-plate) presented in this work (Fig. 2) and provided by **Ref 10, Raufaste et al.** (<https://doi.org/10.1103/PhysRevLett.119.108001>) on superpropulsion (Figure below which we add as **Fig. S10**). The experimental data from Ref. 10 show good matches to the initial and boundary conditions of the CFD model.

Furthermore, based on the initial and boundary conditions of the CFD model, the droplet vibrates within the linear regime (i.e. Weber number $We < 1$), where it undergoes compression followed by an extension. Due to this linearity, any potential nonlinear effects arising in a 3D geometry may be ignored.

5. In the analysis of the critical condition for droplet detachment, a key assumption of the energy conservation model is that the kinetic energy of the droplet exceeds the surface adhesion energy. It is fine to use energy balance to determine the critical condition, but droplet detachment is a local phenomenon, i.e., the surface of the droplet is completely detached from the surface of the stylus. A detachment will occur as long as there is still upward local motion around the detachment area when the surface adhesion energy approaches zero. Because there is strong deformation of the droplet, the velocity within the droplet will be different. Hence, the authors should justify this assumption, for example, by considering how the local phenomenon is connected to the overall energy budget.

We would like to thank the reviewer for highlighting the nuances of droplet detachment due to kinetic energy imparted by the surface of the substrate. In this work, we justify our assumptions in two approaches to examine the effect of surface adhesion on droplet detachment and superpropulsion:

1- Energetic approach

The reason behind the energetic approach is to determine the value of a critical value of (f_o/f) to detach a droplet from a vibrating surface, given a constant contact area and amplitudes. We assume that once the droplet 'dewets' the surface, the droplet is practically detached and not

bound to the surface (free to roll). A similar approach has been considered by **Ref. 29 Boreyko and Chen** (10.1103/PhysRevLett.103.174502) and **Ref. 27 Wang et al.** (doi:10.1017/jfm.2019.576) to calculate the critical velocities and accelerations from textured surfaces. In this work, we simply ignore surface texture then the kinetic energy imparted by the moving substrate will need to overcome the work done by the triple line to dewet the contact area. In addition, although the droplet undergoes deformation, the capillary number $Ca \sim 10^{-1}$, the effect of internal flows is negligible on the droplet dynamics. Changes to the main manuscript **Parahydrophobicity of the anal stylus actuator (page 5)** is included here:

“We assume that droplet ejection occurs when the stylus kinetic energy $E_k \propto 1/2 \rho V (2\pi f A)^2$ (where A is the amplitude of the arclength set by the stylus tip, and V is the volume of the droplet) overcomes the surface adhesion $E_s \propto \gamma(1 + \cos\theta_a)$ (Supplementary Information, Section IV)”

“A more accurate analysis of droplet detachment would include understanding the dynamics of the contact line and shape of the droplets during the vibration cycle.”

2- CFD simulations on droplet detachment

We model the droplet detachment on a vertically vibrating plate for various contact angles. Within the limits of a weber number < 1 (no droplet fragmentation), these simulations do reveal that droplet detachment occurs when the relative velocity between the centroid of the droplet and the plate is positive (upward) when the area between the droplet and the surface approaches zero. We included figure S9 below in supporting information that describes these conditions.

We qualified the statements that we used to define the conditions for droplet detachment and clarified both these conditions in the **main manuscript, supplementary information (CFD section, page 5)**. Also, we explained the nuances of droplet ejections and qualified the statement indicating the importance of understanding the dynamics of the contact line and the

need for further investigation beyond the scope of this paper. We included below the paragraph that we added in the **supplementary information, page 5**:

“Alternatively, CFD simulations of the ejection dynamics of droplets with various contact angles reveal that droplet detachment occurs when the contact line L between the droplet and the substrate approaches zero while the relative speed between the droplet and the substrate is greater than or equal to zero (Fig. S9). An in-depth analysis of superpropulsion and droplet ejection would consider the movement of the contact line and shape analysis of the droplet during ejection.”

6. The caption of Fig. 2d is missing.

Figure 2 is updated, and the caption for fig.2d is included.

7. What is Fig. 1f? The symbol is θ , but the unit is $^\circ/\text{s}$, i.e., for angular velocity.

We thank the reviewer for highlighting this. Fig. 1f is fixed, and the symbol is adjusted to degree/s.

8. How is $t = 0$ defined?

Since it is hard to identify the ‘beginning’ of the movement in the organismal data, we avoid defining $t = 0$ (Unlike the two-spring model and CFD since the functions are prescribed beforehand). The moment $t = 0$ is not used to calculate parameters or draw conclusions.

We provide a discussion in the supplementary information section that clarifies the above information, which we highlight in **the supplementary document (Section II, B)** here:

“This model circumvents the challenge of estimating the beginning and end of the movement. To estimate the effective frequency of the stylus movement, a sinusoidal function $\theta(t) = \theta_0 \sin(2\pi f_s t + \phi)$ may be fitted to the angular speed curve of the raw data..”

How are the max compression time t_c and ejection time t_e defined? Do they mean a period or a moment?

The max compression time t_c and ejection time t_e moments are used instead since they are easier to define than $t = 0$. t_c is defined as when the droplet forms an oblate spheroid with a major axis length that reaches its highest length, and t_e is defined as when the droplet separates from the surface (only in the two-spring model).

We added the following sentence on page 2:

“As a result, the droplets visibly deform into oblate spheroids reaching maximum compression at t_c before extending, supporting elastic energy storage for superpropulsive principles”

What is of interest, however, is the difference between the time the droplet reaches maximum compression and when the substrate reaches maximum speeds i.e. $t_v - t_c$ which is defined in $\tau = (t_c - t_v)/T$.

9. There are many typos in the manuscript. Just to name a few:

- a. In the heading on Page 5, 'Parahydrophibicity' should be 'Parahydrophobicity'**
- b. In the caption of Fig. 3, 'then' should be 'than'.**
- c. In the caption of Fig. 5 in Supplementary Material, 'te' should be 't_e'.**

Thank you for pointing these out. We fixed the typos suggested by the reviewer and proofread the whole manuscript again.

Response to Reviewer #2

We thank Reviewer #2 for the time and effort they spent reviewing our manuscript. In our revisions, we have done our best to carefully consider all of their comments and address all concerns and suggestions which we believe helped us improve the manuscript.

Please find below our point-by-point response. We include two versions of the revised manuscript: an annotated version highlighting all the changes/additions and a non-annotated version. In the revised manuscript, we include a new methods section where we describe the two-spring model and Computational fluid dynamics model, two additional figures in the supplementary information (Fig. S9 and Fig. S10), a new paragraph in the superpropulsion of liquid droplets section, and several minor modifications and additions spread throughout the manuscript.

In the rebuttal document below, the referees' comments are in black, and the author's responses are in blue.

Report on the manuscript « Droplet superpropulsion in an energetically constrained insect »

As stated by the authors, this paper « present the superpropulsion phenomenon for the first time in a biological system » and in that sense it is really interesting and suitable for publication. The paper is well written and several (maybe too much ...) points are addressed (experimental, theoretical and simulations). Nevertheless, following suggestions have to be considered/answered before publication.

1) A much more deeper presentation of the super propulsion phenomenon has to be addressed.

The sentence « Conceptually, this is similar to an Olympic diver perfectly timing their launch off a springboard. » is not appropriate at all ; in that case the energy come from the projectile and not from the catapult. Such a short-cut has to be removed

We thank the reviewer for pointing out the misunderstanding that may have been caused by that statement. The rationale behind using this analogy was to give an intuitive representation of superpropulsion. This statement has been removed in the revision.

and I recommend a much more detailed discussion of the initial papers (ref 10 and 11).

References to :

*** Giombini et al. (Phys Rev E 2022) (spring model, effect of substrate elasticity)**

*** Chantelot et al (Phys. Rev Fluids 2018) (large accelerations)**

*** Celestini et al. (J. Coll Int. Sci 2018) (effect of surface tension and viscosity)**

have to be added to clearly illustrate the discussion/presentation of the superpropulsion phenomenon.

We appreciate the reviewer's insightful suggestion and agree that it would be useful to include more details on superpropulsion. We included a more detailed discussion of the superpropulsion concept in the main text and Methods section and incorporated the references suggested by the reviewer. The changes are highlighted in the annotated manuscript, which we include below:

1- Section: Superpropulsion of liquid droplets

The physics of elastic projectiles had been previously explored in the context of throwing or hitting in ball sports[12, 13], kicking water droplets from superhydrophobic surfaces [10, 14], and propulsion of soft springy materials [11, 15]. One critical feature in elastic propulsion is the importance of sequential timing and synchronization between an actuator and projectile to enhance propulsion by optimizing energy transfer [10, 11, 16].

Superpropulsion is a principle previously described in synthetically engineered systems, where a water droplet may be ejected at a higher speed (up to 1.6 times) than the maximum speed of a superhydrophobic vibrating plate (i.e. $\lambda > 1$) [10]. Such counterintuitive propulsion is achieved only in elastic projectiles (such as a water droplet) by carefully tuning the underlying actuator's vibrational frequency to the projectile's natural frequency. The superpropulsion principle may be exploited to enhance the throwing of rigid projectiles through the elastic tuning offered by compliant substrates [11, 15]. By adding a soft layer with specific geometrical properties to a rigid projectile, the reaction force between the thrower and the projectile might be delayed to increase the initial take-off speed [11]

2- Conclusion and future outlook

In an engineered system, superpropulsion offers a novel means to optimize the catapulting efficiency of rigid projectiles [11]. By adding a soft compliant layer, the actual timescale may be synchronized to the timescale of elastic wave propagation to the projectile [12].

3- Methods section

“We describe the dynamics of the stylus and the droplet with a minimalist mathematical model based on a coupled double mass-spring-damper inspired by the wave model [10], chained spring systems [11, 15] and bouncing model [28] (Fig. S5)”

“...Also since the acceleration of the actuating substrate is relatively low (14 – 28 g, where g is the gravitational constant), the droplet remains in the linear droplet deformation regime and does not form water puddles [9]. Under these conditions, water droplets ($Bo < 1$) deform from a spherical cap into an oblate spheroid and behave as a quasi-Hookean spring with a stiffness $k_d = 32\pi\gamma/3 = 2.41 \text{ N/m}$ ”

2) The two spring model has already been proposed (ref 11 and one of the ref proposed just above) and I think it is better to much more concentrate on CFD simulations that present new results.

We thank the reviewer for pointing this out. Although the previous references present the two-spring model, in our work, we advance the two-spring model to reveal three new insights in the context of biological superpropulsion: dissipation, tuning, and coupling due to stiffness and mass. We expand on each of these below and clarify it in the manuscript and supplementary information:

1- Dissipation: The two-spring model presented in this work examines the effect of dissipation due to losses in the form of adhesion, viscosity, etc., through the damping term - which is previously ignored (**Main text Fig. 2, Supplementary information, Fig. 5**). Furthermore, we are currently exploring the fluid mechanics of feeding and excretion of phloem feeders, which have a significantly different fluidic diet (surface tension, viscosity, etc.) which will be the subject of an ongoing manuscript. In addition, the two-spring model helps strengthen the energetics analysis, mainly in droplet ejection vs jetting, also discussed in **Fig. 4**, and hydrodynamics, further discussed in the **supplementary information, section V on page 5**.

2- Tuning: The two-spring model provides novel insights into temporal tuning in superpropulsion: It reveals the importance of matching by the droplet's maximum compression time and the actuator's maximum velocity-time, highlighted in **Fig. d**. These results complement the CFD simulations and experimental results and provide an alternative way of identifying tuning in a system where the beginning and end of a displacement is hard to define.

3- Coupling: The two-spring model reveals the importance of the stiffness and mass ratio on the coupling dynamics (discussed in the **methods section**) and provides insights into the limits of elastic propulsion in a biological setting (e.g. with resilin as the biological spring). We present this in **figure S8 (shown below)**, where we showcase where a resilin-based spring exploited GWSS couples with the droplet elasticity (f_o/f vs k_d/k_s).

Furthermore, the two-spring model represents the dynamics of a droplet ejecting from a moving substrate which is different from what is presented in **Ref. 11**, where physical masses and springs are modeled. We believe that the model improves the paper's cohesion and appeals to a wider interdisciplinary audience (biophysicists, biologists, ecologists, etc.) where muscles and tendons are often modeled as mass and springs **Ref. 16**. While the CFD simulation captures the fluid dynamics of the droplet (Navier-Stokes), a reduced-order model reduces the complexities of dealing with such equations and presents a simplistic yet powerful means to explain the biological phenomenon. We highlighted this information in the methods section, which we added to the revised manuscript.

3) Ejection experiments of drops standing on a plate have also already been performed in the initial paper presenting superpropulsion (ref 10). I wonder if it is necessary to present these results in this paper. The discussion proposed in point 1) just above will do the job.

The droplet ejection experiments presented here are to validate the computational fluid dynamic simulations, two spring models, and organismal data. The range of f_o/f presented here goes beyond that previously presented in **Ref. 10** (up to $f_o/f = 8$). Including these experiments further validates the concept of superpropulsion and complements the theoretical and computational models. Furthermore, they support several notions discovered in superpropulsion, such as matching the maximum compression and speed times.

4) Figure 2 : the fig 2-d is not commented in the caption. Remove the label d) in the figure.

Fig. 2 is adjusted. We decided to add caption d and adjust captions b and c.

5) The restitution coefficient is defined in terms of velocity. Why the authors did not present it in terms of energy as initially done in ref 10. It will help to compare both experimental and theoretical results.

The restitution coefficient (speed ratio) λ is chosen since it describes the kinematics of the ejection process - which is the core emphasis of this work and directly relates to the initially paradoxical observations of the droplet leaving faster than the speed of the stylus.

We do compare the superpropulsion results of ref. 10 with the two spring-model, CFD and bouncing model in **Fig. S7** in the supporting information, which we include below:

6) Have you measure the surface tension of ejected droplets ? In this paper you actually suppose it is water ... Is this true for excretion of GWSS ?

We did not measure the surface tension of the ejected droplets. However, since it is well known that the sharpshooter's excreted fluid is 99% water **Ref. 5 (Andersen et al.)** and **Ref. 9 (Redak et al.)**, we believe it would be a valid assumption. We clarify this assumption in **Supplementary Information Section V, page 5 of SI**, which we include below:

"Sharpshooter insects have developed unique digestive structures called filter chambers capable of concentrating and separating nutrients with extremely high efficiency to maximize nutrient extraction. The resulting liquid waste is 99% water [23]"

Also, in the **Introduction, page 1** of the main manuscript:

"sharpshooters must constantly and efficiently excrete their fluidic waste (~ 99% water, 'leafhopper rain'), contributing to their role as plant disease vectors [9]."

Response to Reviewer #3

In this original work, the authors report, for the first time, the superpropulsion phenomenon in a biological system. Their study was extremely well performed by studying for example the shape of the droplet or the elasticity of the projectile. I have just some slight comments for this manuscript:

-The term "Parahydrophibicity" should be changed by "Parahydrophobicity". For the parahydrophobicity, the authors should also cite and use this recent work: Experimental characterization of droplet adhesion: the ejection test method (ETM) applied to surfaces with various hydrophobicity, J. Phys. Chem. A, 2018, 122, 8693–8700.

-The authors say "a motif shared by other self-cleaning natural (rose petals) and synthetic substrates." Rose petals do not have self-cleaning properties.

Thank you so much. We adjusted the term "Parahydrophobicity" and cited the suggested work. We also removed the 'self-cleaning' term in this sentence.

REVIEWERS' COMMENTS

Reviewer #1 (Remarks to the Author):

The responses and revisions are satisfactory.